# Effect of Silicon on the Surface Modification of Al-Cr Powder Cathodes Subjected to Vacuum Arc Treatment

Gennady Pribytkov, Victoria Korzhova, Elena Korosteleva and Maksim Krinitcyn *

Institute of Strength Physics and Materials Science SB RAS, Tomsk 634055, Russia; gapribyt@mail.ru (G.P.); vicvic5@mail.ru (V.K.); elenak@ispms.ru (E.K.)
* Correspondence: krinmax@gmail.com

**Abstract:** Al-Cr and Al-Cr-Si composite cathodes were obtained by the hot compaction of aluminum, chromium, and silicon powder mixtures. The phase transformations in the surface layer of the Al-Cr-Si composite cathodes subjected to the arc heating were considered. The elemental and phase compositions of the modified cathodes' surfaces were studied using X-ray diffraction, scanning electron microscopy (SEM), and energy dispersive X-ray spectroscopy (EDS). The effect of the silicon addition on the structural evolution in the cathode surface during arc evaporation is shown. It was found that an arc impact on the cathode surface resulted in the melting and consequent crystallization of the multiphase mixture of intermetallic compounds and eutectic in the cathode surface layer. Cathode surface layers were found to be depleted of aluminum and silicon due to the ejection of these elements in drop form from the Al-Si liquid layer on the cathode surface. This can result in the change in the elements ratio in the deposited coating as compared with that in the cathode and thus influence the coated tools' durability.

**Keywords:** Al-Cr-Si powders; hot compaction; composite cathodes; cathodic arc deposition; surface melting; arc erosion; element depletion

## 1. Introduction

Wear-resistant coatings for metalworking tools increase a tool life due to their high hardness, resistance to air oxidation, and a reduced coefficient of friction on the material being worked. Coatings are applied to the tool surface by vacuum arc evaporation or magnetron sputtering of cathodes in reactive gases, mainly in nitrogen and oxygen [1,2]. When a titanium cathode is exposed to vacuum arc evaporation in nitrogen gas, a thin film of titanium nitride is deposited on the tool surface. The elemental composition of the titanium cathodes has been doped by other nitride-forming elements (aluminum, chromium, molybdenum, and silicon). The greatest increase in tool life was obtained on tools with TiAlN, CrAlN, and CrAlSiN nitride coatings [3–5]. AlCrN and AlCrSiN coatings on metalworking tools have excellent thermal stability, air-oxidation resistance, and a reduced coefficient of dry friction against steels [6–9]. During vacuum arc evaporation, the cathode material is ejected from vacuum arc craters on the cathode surface in ion form with different charges, neutral atoms, and melted drops [10,11]. In the investigations devoted to cathodic arc-coating technologies, attention is paid to the deposition process features and to the coatings' structure and properties [12,13]. The effect of the cathode's elemental composition and the cathode's evaporation modes on the coated tool durability are also examined [14,15]. Lttle attention is paid to the cathode surface, which is a source of plasma flow generated by the arc.

It is known that the surfaces of cathodes, used for coating deposition by vacuum arc evaporation, are heated to high temperatures exceeding the melting point of the cathode material. Surface melting results in an increased number of drops in the products of the arc erosion. The greater the drop quantity, the lower the melting temperature of the

cathode [16]. Heating of the surface layer of multicomponent cathodes, which are in a nonequilibrium state, for example, consisting of a mechanical mixture of elementary powders, is accompanied by structural and phase transformations, which can change the ejection properties of the surface layer of cathodes. In particular, on the surface of cathodes composed of a compacted powder mixture of aluminum (50 at.%) and titanium subjected to vacuum arc heating, the formation of titanium aluminides occurred [17]. Titanium aluminides in a surface layer of the vacuum-arc-powder cathodes were also detected in [18]. The origin of the aluminides was not discussed. The coatings deposited by the cathodic arc evaporation had high roughness because of the solidified drops of the plasma jets [19–21]. The Al-Cr and Al-Cr-Si cathodes, hot compacted from powder mixtures, contain a low-melting aluminum. Therefore, the drop content in the plasma increases, resulting in surface quality deterioration as a result of the drops' crystallization on the coating surface. There are studies [19] where analysis results of the arc impact on the elemental and phase state of Al-Cr cathodes under various gas media are presented, but a detailed assessment of the element balance during the arc evaporation has not been provided. According to [19–22], an Al-Cr intermetallic formation occurred on the cathodes' surface consisting of the compacted powder mixture of aluminum and chromium, as a result of the arc impact. Here too, the origin of the Al-Cr aluminides was not discussed. There was also no information about the silicon effect on erosive processes in these cathodes during the plasma generation. Thus, it is of great interest to investigate the microstructure, the phase, and the elemental composition of the surface layer of multicomponent cathodes modified by the arc impact. The investigation results can clarify the influence of the cathode surface layer on the coatings' elemental composition and properties, as well as the effect on tool durability in machining operations.

The mechanical properties of the hot compacted Al-Cr and Al-Cr-Si powder materials were investigated previously [23]. According to our results, the residual porosity, strength, and ductility of the hot compacted cathode was appropriate for the use condition of the vacuum arc and magnetron cathodes.

## 2. Materials and Methods

The targets for the cathodic arc deposition were obtained by hot (550 °C) compaction (0.5 GPa) of the preforms produced by cold consolidation of the powder mixtures of aluminum ($\leqslant$80 μm), chromium ($\leqslant$50 μm), and silicon ($\leqslant$80 μm). The powder shape is presented in Figure 1.

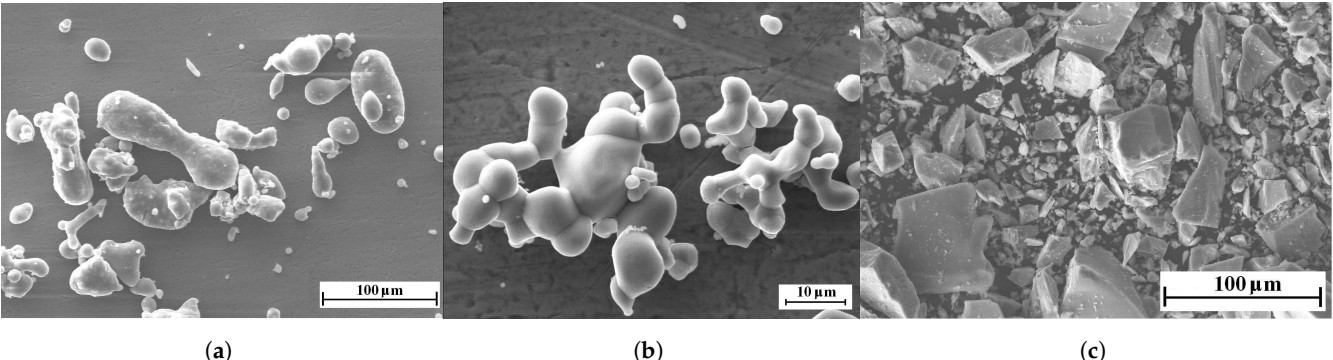

| (a) | (b) | (c) |

**Figure 1.** SEM images of the powders used for the cathode compactions: (**a**) Al; (**b**) Cr; (**c**) Si.

The cathodes of $Al_{0.7}Cr_{0.3}$, $Al_{0.65}Cr_{0.25}Si_{0.1}$, and $Al_{0.6}Cr_{0.2}Si_{0.2}$ (at.%) compositions were investigated. The coatings were deposited in $N_2$ gas under 0.065 Pa pressure and a 90 A arc current for 60 min.

The phase composition was determined by X-ray diffraction using Co K$\alpha$ radiation. The X-ray data of the ASTM data file and the PDWin (4.0, NPP Bourevestnik, Saint-Petersburg, Russia) software were used for phase identification and characterization. Structure studies of the cathodes were carried out by optical (AXIOVERT-200MAT) and scanning

electron (LEO EVO 50, Zeiss, Germany) microscopy. The elemental composition of the modified layer was determined by X-ray spectroscopy (EDS).

## 3. Results and Discussion

The microstructure of the hot-compacted powder cathodes represented an aluminum matrix with inclusions of chromium (Figure 2a) or of chromium and silicon particles (Figure 2b). The porosity of the hot compacted cathode material was near to zero.

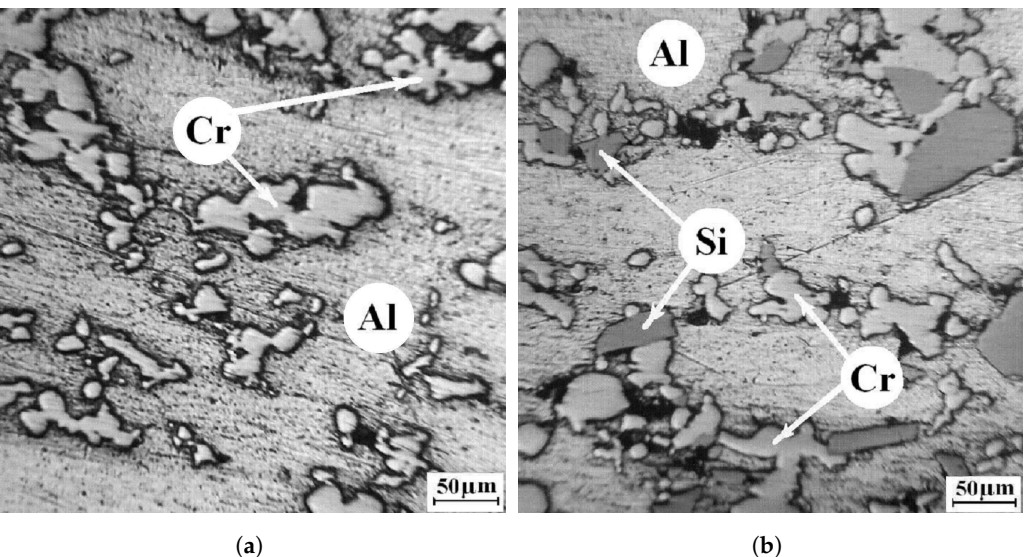

(**a**)          (**b**)

**Figure 2.** Optical microscopy images of the cathode structure: (**a**) $Al_{0.7}Cr_{0.3}$; (**b**) $Al_{0.65}Cr_{0.25}Si_{0.1}$.

Figure 3 shows top views of the cathode surfaces after the arc treatment. The smooth view of the surfaces was evidence of the surface fusion of the cathode material. A crack formation due to the crystallization of the melt film can be seen on the surface. A network of cracks on the surfaces was an indication of the low plasticity of the solidified liquid surface layer.

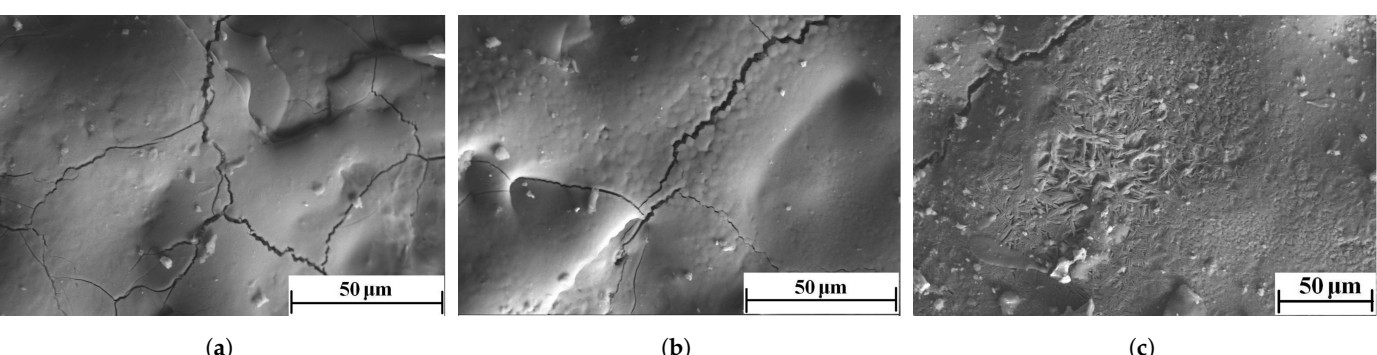

(**a**)          (**b**)          (**c**)

**Figure 3.** SEM images (top view) of the arc-treated cathodes: (**a**) $Al_{0.7}Cr_{0.3}$; (**b**) $Al_{0.65}Cr_{0.25}Si_{0.1}$; (**c**) $Al_{0.6}Cr_{0.2}Si_{0.2}$.

Figure 4 shows cross sections of the modified layers. The thickness of the modified layer depended on the silicon content in the cathodes and increased from 5–30 μm on the $Al_{0.7}Cr_{0.3}$ to 80 μm on the $Al_{0.65}Cr_{0.25}Si_{0.1}$ cathode. One reason for the wide range in the layer thicknesses was the inhomogeneous distribution of the Cr and Si particles in the Al matrix resulting in a wide variation in the depth of the melted layer. The transition zone between the modified layer and the cathode body was not observed. That is, no reactive solid-state diffusion occurred. So, the crystallization origin of the modified layer was proved.

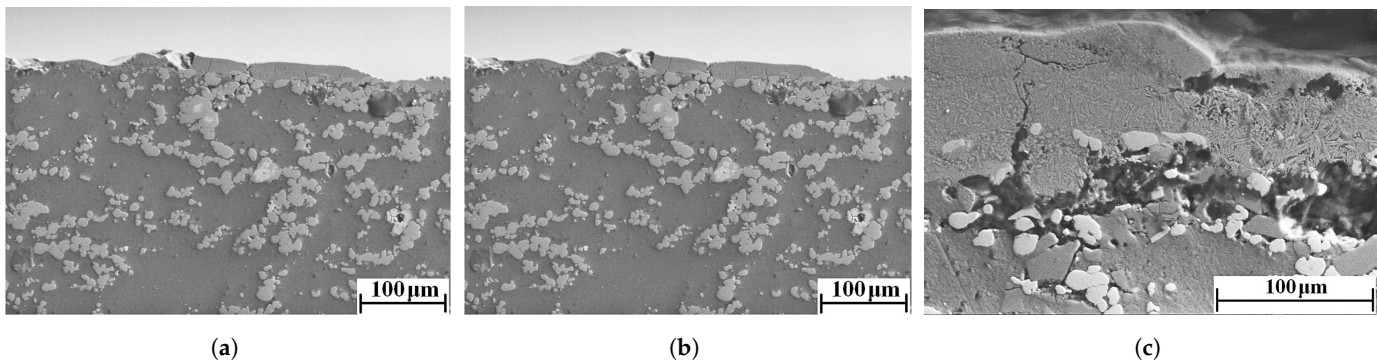

|       |       |       |
| :---: | :---: | :---: |
| (**a**) | (**b**) | (**c**) |

**Figure 4.** SEM image of the cross section of the cathodes' modified layer: (**a**) $Al_{0.7}Cr_{0.3}$; (**b**) $Al_{0.65}Cr_{0.25}Si_{0.1}$; (**c**) $Al_{0.6}Cr_{0.2}Si_{0.2}$.

The thickness of the modified surface layer on the $Al_{0.6}Cr_{0.2}Si_{0.2}$ cathode increased to 120 µm (Figure 4c). In this case, the multiphase composition of the modified layer presented along with the typical transverse cracks. At the boundary between the modified layer and the cathode bulk, cavities appeared due to the crystallization shrinkage. A significant part of this layer was occupied by a crystallized Al-Si eutectic. The large volume of low-melting eutectic is explained by the double silicon content in the mechanical mixture of powders as compared to the $Al_{0.65}Cr_{0.25}Si_{0.1}$ composition. Therefore, the number of contacts of adjacent particles of aluminum and silicon, on which a eutectic liquid appeared, increased. The process of contact melting was accelerated, and the volume of the resulting eutectic liquid increased. The surface layer was separated from the initial material of the cathode by a sharp boundary, on which there were defects in the form of cavities formed during the crystallization of the melt.

The change in the phase composition on the cathode working surface was confirmed by the results of the X-ray diffraction analysis (Figure 5, Table 1). The top of the cathode surface was X-ray exposed. The multiphase composition of the modified layers was the result of the nonequilibrium conditions of the melting and crystallization. The XRD data showed that in the Al-Cr cathode surface layer, the main phase was $Al_8Cr_5$. Silicon addition up to 20 at.% led to the $Al_{74}Cr_{20}Si_6$ ternary phase formation along with $Al_8Cr_5$. The aluminum content in the $Al_8Cr_5$ was more than that in the cathode body. Free aluminum and silicon belonged to the eutectic in the modified layer of the $Al_{0.6}Cr_{0.2}Si_{0.2}$ cathode (Table 1).

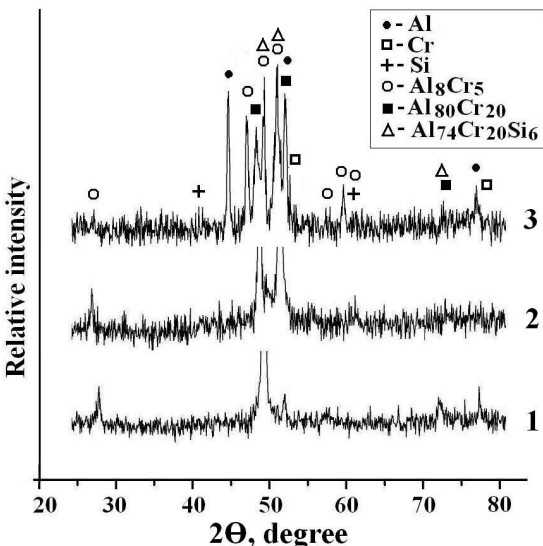

**Figure 5.** XRD patterns from the top of the modified layer of the cathodes: (**1**) $Al_{0.7}Cr_{0.3}$; (**2**) $Al_{0.65}Cr_{0.25}Si_{0.1}$; (**3**) $Al_{0.6}Cr_{0.2}Si_{0.2}$.

**Table 1.** Phase composition of the modified layer (vol.%).

| Cathode | Al | Cr | Si | $Al_4Cr$ | $Al_{74}Cr_{20}Si_6$ | $Al_8Cr_5$ |
|---|---|---|---|---|---|---|
| $Al_{0.7}Cr_{0.3}$ | 16.7 | 9.8 | - | 12.4 | - | 61.1 |
| $Al_{0.65}Cr_{0.25}Si_{0.1}$ | 10.2 | 8.9 | 5.2 | 17.4 | 21.6 | 36.7 |
| $Al_{0.6}Cr_{0.2}Si_{0.2}$ | 25.2 | 5.4 | 13.6 | 16.5 | 13.5 | 25.8 |

The existence of a continuous melt film on the surface of the Al-Cr and Al-Cr-Si powder cathodes under the conditions of stationary vacuum arc evaporation resulted in enhanced droplet generation. The formation of the $Al_8Cr_5$ intermetallic in the modified surface layer of the $Al_{0.7}Cr_{0.3}$ cathode was an effect of the crystallization of the aluminum-depleted melt but not a consequence of the aluminum and chromium reaction interdiffusion. The melting of the cathode surface was promoted by a high content of low-melting aluminum (76.4 vol.%) in the mechanical mixture with chromium. The addition of silicon increased the thickness of the melted layer on the cathode surface due to the formation of low-melting Al-Si eutectic.

An additional factor contributing to the formation of the intermetallic compounds by reaction diffusion in elemental powder mixtures was a large interfacial reaction surface, that is, the use of the finest powders. However, more research is required with a variety of powder mixture compositions, powder dispersion, and arc evaporation modes, primarily arc current, to elucidate the possibility and conditions for the formation of high-melting intermetallic layers on the working surface of powder cathodes under arc treatment.

The elemental composition of the modified layer was determined by energy dispersive X-ray spectroscopy (EDS); the distribution of aluminum and chromium in the volume of the $Al_{0.7}Cr_{0.3}$ cathode and in the modified layer is clearly visible in the map images (Figure 6).

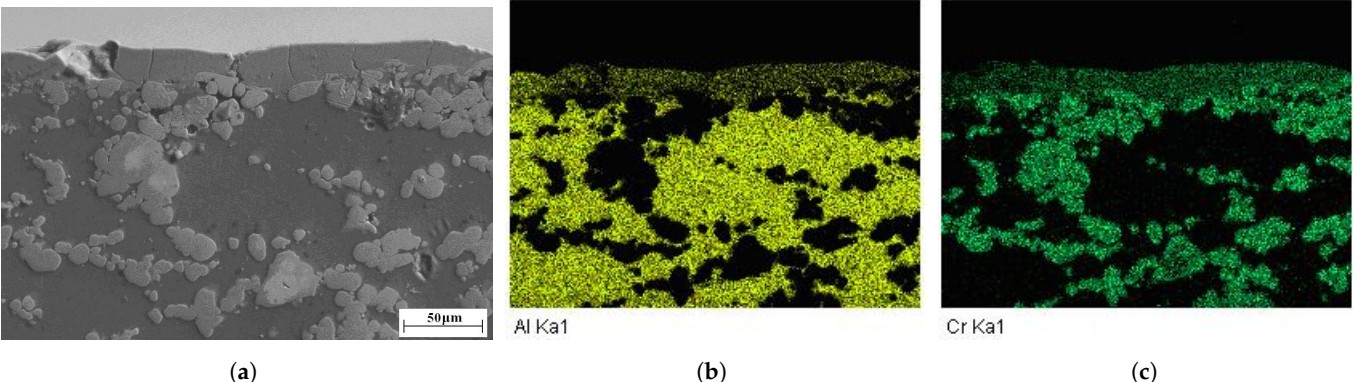

(**a**)　　　　　　　　　　　(**b**)　　　　　　　　　　　(**c**)

**Figure 6.** EDS element (Al (**b**) and Cr (**c**) elements) mapping in the near-surface region (**a**) of the $Al_{0.7}Cr_{0.3}$ cathode according to the results of the EDS analysis.

The elemental composition of the cathode modified layer was investigated both on the top surface and on cross sections (Figure 7). The averaged results of the EDS analysis in the counted areas on Figure 7 are presented in Table 2. The Al/Cr ratio (1.55) in the top part of the modified layer on the $Al_{0.7}Cr_{0.3}$ cathode was much lower than that (2.1) in the bottom part adjacent to the cathode body (Tables 2 and 3). Both values were different from the initial cathode Al/Cr ratio (2.33). That means there was a depletion of the modified layer with aluminum as a result of its predominant ejection in drop form under the arc impact. In this case, the Al/Cr ratio (1.55) approximately corresponded to the $Al_8Cr_5$ phase (Al/Cr = 1.6) that was the dominant phase in the thin surface layer of the $Al_{0.7}Cr_{0.3}$ cathode according to the XRD results.

**Table 2.** The distribution of the elements in the modified layer (top and cross section).

| Cathode | | C | N | O | Al | Cr | Si |
|---|---|---|---|---|---|---|---|
| $Al_{0.7}Cr_{0.3}$ | Top | $12.8 \pm 6.6$ | $4.8 \pm 1.4$ | $1.7 \pm 0.6$ | $49.5 \pm 5.4$ | $31.2 \pm 4.0$ | - |
| | Cross section | $25.2 \pm 11.1$ | $1.3 \pm 1.4$ | $1.1 \pm 0.8$ | $49.1 \pm 6.6$ | $23.3 \pm 5.8$ | - |
| $Al_{0.65}Cr_{0.25}Si_{0.1}$ | Top (int) | $17.4$ | $2.5$ | $2.4$ | $47.9$ | $22.8$ | $7.0$ |
| | Cross section | $17.9 \pm 13.4$ | $1.7 \pm 1.0$ | $1.0 \pm 1.2$ | $51.9 \pm 10.5$ | $20.0 \pm 4.3$ | $7.5 \pm 1.2$ |
| $Al_{0.6}Cr_{0.2}Si_{0.2}$ | Top | $34.1 \pm 18.4$ | $4.9 \pm 2.8$ | $10.7 \pm 10.0$ | $25.8 \pm 16.7$ | $9.2 \pm 6.6$ | $15.3 \pm 11.6$ |
| | Cross section | $34.0 \pm 22.7$ | $0.7 \pm 0.8$ | $0.8 \pm 1.3$ | $39.7 \pm 15.1$ | $12.9 \pm 4.6$ | $11.9 \pm 3.8$ |

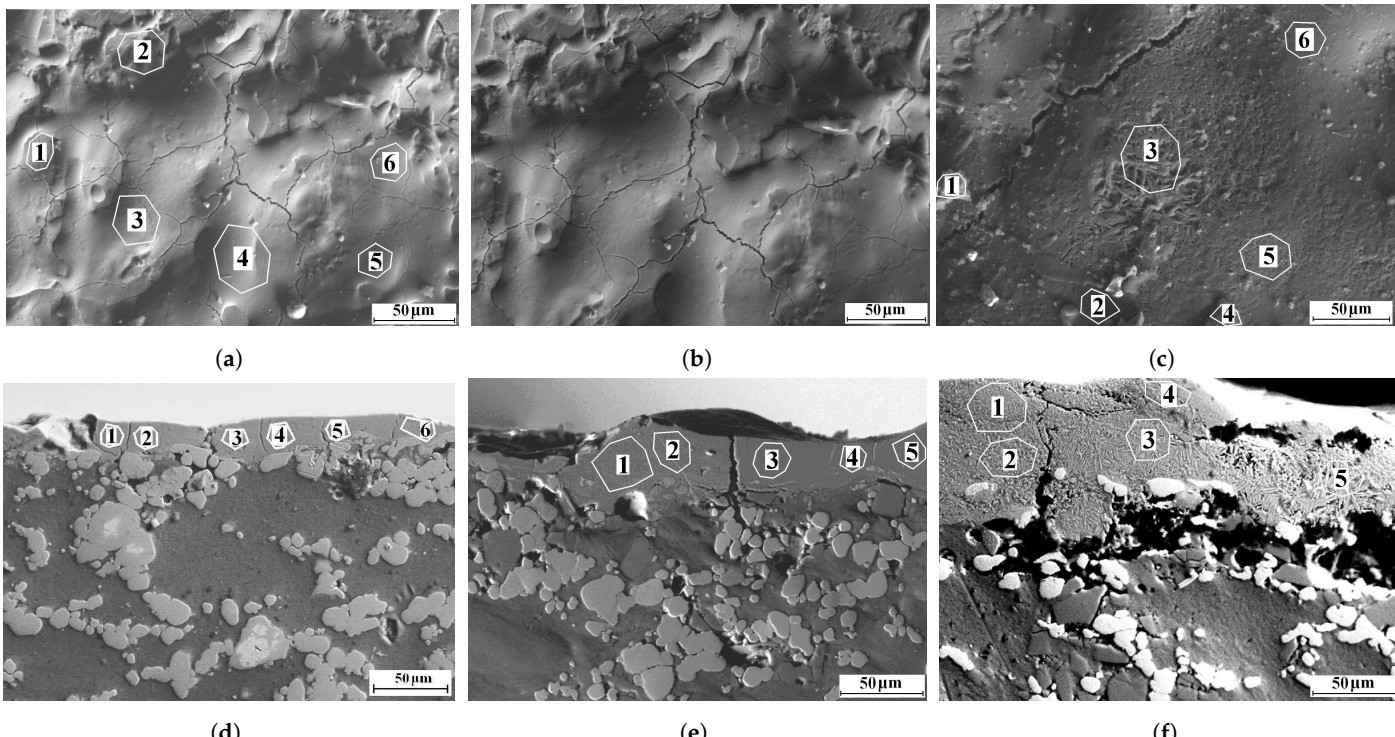

(a)　　　　　　　　　　(b)　　　　　　　　　　(c)

(d)　　　　　　　　　　(e)　　　　　　　　　　(f)

**Figure 7.** Elemental analysis of the modified layer of the cathodes: (**a**,**d**) $Al_{0.7}Cr_{0.3}$; (**b**,**e**) $Al_{0.65}Cr_{0.25}Si_{0.1}$; (**c**,**f**) $Al_{0.6}Cr_{0.2}Si_{0.2}$. (**a**–**c**)—top view; (**d**–**f**)—cross section. Investigated areas are marked with numbers.

**Table 3.** Elements ratio in the top and cross section parts of the modified layer.

| Area | Elements Ratio | Cathode | | |
|---|---|---|---|---|
| | | $Al_{0.7}Cr_{0.3}$ | $Al_{0.65}Cr_{0.25}Si_{0.1}$ | $Al_{0.6}Cr_{0.2}Si_{0.2}$ |
| Top of | Al/Cr | 1.55 | 2.10 | 2.81 |
| modified layer | Al/Si | - | 6.84 | 3.33 |
| | Cr/Si | - | 3.26 | 1.18 |
| Cross-section of | Al/Cr | 2.10 | 2.60 | 3.05 |
| modified layer | Al/Si | - | 6.83 | 3.41 |
| | Cr/Si | - | 2.63 | 1.12 |

A similar effect of aluminum depletion was also observed for the $Al_{0.65}Cr_{0.25}Si_{0.1}$ cathodes. The Al/Cr ratio (2.10) for the top surface was less than that for the cathode bulk (2.60). According to the EDS analysis, the Al/Cr ratio became equal to that for the cathode bulk when moving away from the top surface to the bottom of the modified layer. The Al depletion effect in the modified $Al_{0.65}Cr_{0.25}Si_{0.1}$ cathode surface occurred only in a thin layer ($\approx 5$ μm).

　　　　Along with Al depletion, Si depletion occurred in the top thin layer of the cathode. Apparently, this depletion of silicon was also related to the silicon ejection in the drops of low-melting Al-Si eutectic. The difference between the Al/Si ratios for the top and bottom of the modified layer was minor in both the $Al_{0.65}Cr_{0.25}Si_{0.1}$ and $Al_{0.6}Cr_{0.2}Si_{0.2}$ cathodes. The reason is the intensive convective mixing in the molten surface film that was thicker on the surface of the Si-containing cathodes than that on the Al-Cr cathode surface.

## 4. Conclusions

　　　　The surface layers of the powder Al-Cr-Si hot-compacted cathodes subjected to arc heating were melted up to 120 μm in depth, depending on the elemental composition of the cathode. After cooling, the melted layer solidified, forming a multiphase mixture of intermetallic compounds and eutectic. The silicon addition to the Al-Cr powder mixtures resulted in a thickness increase of the melted layer on the cathode surface. A novelty of the work lies in the fact that in the stationary mode of the arc evaporation, the aluminum and silicon depletion of the surface layer occurred due to their ejection in drop form from the Al-Si liquid film on the cathode surface. The depletion can change the elements ratio in the coating deposited.

**Author Contributions:** Conceptualization and experimental design, G.P. and V.K.; methodology, and analysis and discussion of the results, E.K.; EDS analysis and the manuscript writing, M.K. All authors have read and agreed to the published version of the manuscript.

**Funding:** The work was performed according to the Government research assignment for ISPMS SB RAS, project FWRW-2021-0005.

**Data Availability Statement:** The data presented in this study are available on request from the corresponding author.

**Conflicts of Interest:** The authors declare no conflict of interest.

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
