# Peer review of "Effect of Silicon on the Surface Modification of Al-Cr Powder Cathodes Subjected to Vacuum Arc Treatment"

_coatings, doi:10.3390/coatings12070958_

Round 1

Reviewer 1 Report

Review report on the topic ‘Effect of silicon on the surface modification of Al-Cr powder cathodes subjected to vacuum arc treatment’. Comments are listed below:

1.       Strengthen the abstract section. Add the key conclusion of the works in the last two lines of the abstract section.

2.       Discuss the motive behind the work. The clear application of the work should be discussed in the introduction section. From the introduction section application of the work is not clear.

3.       There are numerous spelling and grammatical errors. Please revise the manuscript thoroughly. Sentences are also not complete.

4.       The novelty of the work should also be discussed in a separate paragraph.

5.       Try to make a bridge between current and previously published work and specify the gap area and objective of the work. Add the specific gap observed from the literature at the end of the introduction section. Refer to the following: https://doi.org/10.1007/s12540-020-00705-w; https://doi.org/10.1016/j.ceramint.2018.01.131.

6.       Provide more detail about the experimental section.

7.       Interface surface needs more characterization.

8.       XRD results need more technical discussion and

9.       The manuscript is written very poorly. A lot of English and grammatical errors in the manuscript. Please improve the quality of the writing.

Author Response

Dear Reviewer, Thank you very much for your attention to our paper. Your comments and remarks were very usefully for us. We tried to answer your questions as fully as possible. 1.    The sentences concerning to the objective and practical importance of the work were added to the abstract (green marked).
2.    Some elucidation to the research motivation was presented in the introduction section (green marked).
3.    The detected errors were corrected
4.    An addition concerning to the novelty of the work is introduced into the conclusion section (green marked).
5.    Some sentences for better understanding  of the work object were introduced into the text. We consider nevertheless that the references presented in the paper are sufficient as a background for the work.
6.    Two papers recommended by the reviewer concern with thermal barrier coatings and are not related to the work topic. 
7.    X-ray diffraction features are added to the Experimental section
8.    Additional description of the “modified layer – cathode body” interface was represented (green marked)
9.    Some details were added to XRD results discussion (green marked).
10.    The detected errors were corrected

Reviewer 2 Report

The manuscript, entitled Effect of silicon on the surface modification of Al-Cr powder cathodes subjected to vacuum arc treatment is relevant to the scope of this journal. It is an interesting study that can bring valuable information to specialists.

Therefore, the article can be recommended for publication only after revision according to the following suggestions:

1.     Bibliographical references should be enhanced with other information from the literature.

2.     The images presented in Figures 1, 2 and 3 are SEM or optical microscopy images? Authors should specify this in the text and Figure captions.

3.     In Figure 3, the insertion of images obtained at higher magnification would be indicated!

4.     Figure 4 should indicate the limits for measuring the thickness or highlight the layer whose thickness has been measured. Also, the layer does not seem to have a uniform thickness for each case, so the thickness values must be presented with errors.

5.     “Table ??” appears on lines 96 and 99. Please correct and add the number of Tables.

6.     How was the phase identification from the XRD spectra made? Cards, software or bibliographic references must be specified!

7.     What is the difference between image a in Figure 6 and image d in Figure 7? It's the same?

8.     It should be noted in the caption of Figure 6 that it is a mapping of the elements.

9.     Some dots are marked in the images in Figure 7. What do they represent? Where do we find them in tables or explanations?

10.  The conclusions are too brief. A few more significant remarks need to be added.

11.  The way of writing the authors' contribution is not according to the requirements of the journal. Please correct.

12.  The comparison of the obtained results with other similar ones existing in the literature is very important and must be done.

Author Response

Dear Reviewer, Thank you very much for your attention to our paper. Your comments and remarks were very usefully for us. We tried to answer your questions as fully as possible. 1.    Authors consider that references presented are sufficient as a background for the work.
2.    The figures captions were corrected
3.    The corrections were done.
4.    Some additional considerations concerning to the modified layer thickness are presented.
5.    The corrections were done.
6.    X-ray diffraction features are added to the Experimental section.
7.    It's the same image. Image in the figure 6 demonstrates a distribution of Al and Cr over the layer while the legends on the image 7d indicate the areas of EDS analysis. The averaged results of EDS analysis are presented in table 2. 
8.    The figure 6 caption was corrected.
9.    In the table 2 averaged results of EDS analysis in the countered areas on the top and cross-section of the modified layer in the figure 7 are presented. A proper interpretation in the text are marked.
10.    A sentence concerning to the work novelty was added to the conclusions.
11.    The corrections were introduced to the text.
12.    We did not find any information on the elemental composition of the modified surface layer. So evidences of the depletion of the surface layer with light elements under vacuum arc action are novel.

Reviewer 3 Report

The manuscript deals with a subject that is relevant to Coatings but in my opinion certain improvement is necessary.

The main concern is regarding the results and discussion section. Obtained results are not put in the context of results by other researchers so the contribution of this article is not clear. Please improve the discussion accordingly.

In addition, some of the research presented in this manuscript has already been published before but authors have missed to properly cite their own reference (Korosteleva, E.N.; Pribytkov, G.A.; Korzhova, V.V. Effect of the Hot Deformation Conditions on Structure and Mechanical Properties of AlCr/AlCrSi Powder Composites. Metals 2021, 11, 1853. https://doi.org/10.3390/ met11111853) for results in figure 3.

Figure 7- EDS is conducted in selected points but the point numbers are not presented in relevant Tables.

XRD measurements- it is not clear in which shape were samples for XRD ?

Line 18- „When titanium cathode is exposed to vacuum-arc evaporation in the nitrogen gas. A thin film of titanium nitride is deposited on the tool surface „ Please merge this two sentences into one.

Line 96, 99- Table numbers are missing

Author Response

1.    We failed to find in the literature the results concerning to the elemental distribution over the cathode layer, modified with the vacuum arc. So evidences of the depletion of the surface layer with light elements under vacuum arc action are novel.
2.    The referred paper concerns to the mechanical properties of the cathodes and do not relates to the work topic. 
3.    The legends on the image 7d indicate the areas of EDS analysis. The averaged results of EDS analysis are presented in table 2. The proper elucidation is given in the text.
4.    X-ray diffraction patterns were taken from the top cathode surface . The proper elucidation is given in the text.
5.    The mistake is corrected. Thanks.
6.    The mistake is corrected. Thanks

Round 2

Reviewer 1 Report

Accepted.

Author Response

Dear reviewer!
Thank you for your appreciation of our work! Thanks to your efforts, the article became clearer for readers and more fully revealed the idea of our research.

Reviewer 2 Report

The authors have made all the required corrections and additions, so that the manuscript now has a much improved form and can be considered for publication.

Author Response

Dear reviewer, thank you for your appreciation of our work! Thanks to your efforts, the article became clearer for readers and more fully revealed the idea of our research.

Reviewer 3 Report

The manuscript is well revised except for the fact that authors didn't cite the original source of Figure 3 which is already presented in their previous paper. In my belief if the same image is repeated the original work must be cited.

Author Response

Dear Reviewer! Thank you for your reply! Our previous result concerning the mechanical properties of the hot compacted powder materials are cited (No. 23 in the References). Appropriate comments are added in the Introduction section.